# Polygenic scores for smoking and educational attainment have independent influences on academic success and adjustment in adolescence and educational attainment in adulthood

Brian M. Hicks [1]*, D. Angus Clark[1], Joseph D. Deak[2,3], Jonathan D. Schaefer [4], Mengzhen Liu[5], Seonkyeong Jang[5], C. Emily Durbin[6], Wendy Johnson[7], Sylia Wilson[4], William G. Iacono[5], Matt McGue[5], Scott I. Vrieze[5]

1 Department of Psychiatry, University of Michigan, Ann Arbor, MI, United States of America, 2 Department of Psychiatry, Yale University, New Haven, Connecticut, United States of America, 3 Department of Psychiatry, Veterans Affairs Connecticut Healthcare Center, Connecticut, United States of America, 4 Institute of Child Development, University of Minnesota, Minneapolis, Minnesota, United States of America, 5 Department of Psychology, University of Minnesota, Minneapolis, Minnesota, United States of America, 6 Department of Psychology, Michigan State University, East Lansing, Michigan, United States of America, 7 Department of Psychology, University of Edinburgh, Edinburgh, Scotland

* brianhic@med.umich.edu

## Abstract

Educational success is associated with greater quality of life and depends, in part, on heritable cognitive and non-cognitive traits. We used polygenic scores (PGS) for smoking and educational attainment to examine different genetic influences on facets of academic adjustment in adolescence and educational attainment in adulthood. PGSs were calculated for participants of the Minnesota Twin Family Study ($N = 3225$) and included as predictors of grades, academic motivation, and discipline problems at ages 11, 14, and 17 years-old, cigarettes per day from ages 14 to 24 years old, and educational attainment in adulthood (mean age 29.4 years). Smoking and educational attainment PGSs had significant incremental associations with each academic variable and cigarettes per day. About half of the adjusted effects of the smoking and education PGSs on educational attainment in adulthood were mediated by the academic variables in adolescence. Cigarettes per day from ages 14 to 24 years old did not account for the effect of the smoking PGS on educational attainment, suggesting the smoking PGS indexes genetic influences related to general behavioral disinhibition. In sum, distinct genetic influences measured by the smoking and educational attainment PGSs contribute to academic adjustment in adolescence and educational attainment in adulthood.

**Data Availability Statement:** The data are available in files posted on the Open Science Framework (https://osf.io/92esr/).

**Funding:** This work was supported by United States Public Health Service grants R37 AA009367 (McGue), R01 AA024433 (Hicks), R21 AA026632 (Wilson), T32 AA028259 (Vasiliou), and T32 AA007477 (Blow) from the National Institute of Alcohol Abuse and Alcoholism, T32 MH015755 (Cicchetti) from the National Institute of Mental Health, and R01 DA034606 (Hicks), R37 DA005147 (Iacono), R01 DA013240 (Iacono), R01 DA044283 (Vrieze), R01 DA037904 (Vrieze), and U01 DA046413 (Vrieze) from the National Institute on Drug Abuse.The funders had no role in study design, data collection and analysis, decision to publish, or preparation of the manuscript.

**Competing interests:** The authors have declared that no competing interests exist.

# Introduction

Educational success is important for a variety of important life outcomes including wealth accumulation, health and longevity, and happiness [1–3]. Success in school entails multiple facets including motivation and enthusiasm for striving for academic goals, willingness to conform to school's standards of conduct, and earning good grades. Consequently, educational success is complex and calls upon a variety of cognitive and non-cognitive traits including intellectual abilities (learning, memory, reasoning), persistence in pursuit of long-term goals, positive activation for goal striving, self-control, and internalizing the importance of academic goals and cultivating positive relationships with adults in education settings [4–6].

Meta-analyses of twin studies have found that genetic influences account for about 40% while shared and nonshared environmental influences each account for about 30% of the variation in educational attainment [7, 8]. These cumulative genetic and environmental influences are observed earlier in development on intermediate phenotypes associated with academic success including grades, academic motivation, and disciplinary conformity, though relative proportions of genetic and environmental influence may differ across these domains and with age [9, 10]. Large genome-wide association study (GWAS) meta-analyses of educational attainment have now identified over 1200 genome-wide significant associations ($p < 5.0$ x $10^{-8}$) with individual single nucleotide polymorphisms (SNPs), a key step in delineating biological processes that contribute to educational attainment [11]. Because effect sizes for individual SNPs are typically very small, polygenic scores (PGS) are often used to aggregate the effects of all SNPs from a GWAS [12]. By weighing all SNPs according to their effect sizes in GWAS, PGS account for about 10% of the variance in educational attainment. PGS then can be used to examine the genetic associations between educational attainment and its known correlates.

Smoking is a non-cognitive trait that has a strong association with lower educational attainment [13, 14]. Rather than a direct causal effect of education on smoking (or vice versa), however, there has been a long recognition that this association is due to the common influences of third variables. For example, smoking patterns tend to be established in the late teens to early 20's, which is prior to the completion of higher education but later than when consistent individual differences in factors strongly related to educational attainment (e.g., GPA, academic motivation, discipline problems) have emerged [15]. Further, sib-pair difference analyses have found that familial factors account for the association between smoking and educational attainment [16]. Finally, recent GWAS findings have estimated genetic correlations from $r$ = .27 to .56 between smoking and educational attainment phenotypes, indicating at least some of their familial association is due to common genetic influences [17, 18].

Behavioral disinhibition refers to difficulty inhibiting impulses to behave in socially undesirable or restricted actions [19] and is a another non-cognitive trait that has been associated with academic success [4]. Externalizing problems are manifestations of these poor inhibitory abilities and include impulsivity, aggression, rule breaking, oppositionality, hyperactivity, and inattention. They are associated with lower grades, poor academic motivation, and more disciplinary problems, and predict lower educational attainment [20, 21], with most of the overlap attributable to shared genetic influences [9]. Smoking, especially in adolescence, is strongly correlated with externalizing behaviors, alcohol use, and other drug use, all of which are manifestations of a higher-order behavioral disinhibition trait [19, 22–24]. It is possible then that the association between smoking and education attainment is actually due to the overlap between smoking and the broader trait of behavioral disinhibition.

Recently, we examined the predictive validity of a PGS for having ever been a regular smoker that was derived from the largest GWAS of smoking-related phenotypes to date

($N$ = 1,232,091) [25]. In replication samples, this PGS accounted for 4% of the variance in a similar smoking phenotype and was also significantly associated with use measures of alcohol, cannabis, cocaine, amphetamines, ecstasy, and hallucinogens [25, 26]. Using the same twin sample as in this report, we found that this smoking PGS predicted trajectories of nicotine and alcohol use from ages 14 to 34, even after adjusting for nicotine and alcohol use and a PGS for drinks per week [27].

This smoking PGS was also associated with the externalizing dimension of the Child Behavior Checklist in a large sample of pre-adolescents, even after adjusting for a general factor of psychopathology [28]. We followed up these results and found that the smoking PGS was associated with externalizing problems and personality traits associated with behavioral control—but not internalizing problems and extraversion—from ages 11 to 17 [29]. We concluded that the smoking PGS was also a measure of genetic influences on general behavioral disinhibition rather than smoking or nicotine addiction specifically, and so could be used to investigate the role that genetic influences related to behavioral disinhibition have on the development of other near-neighbor outcomes.

Here, we examined the relative effects of PGSs for educational attainment and smoking on educational attainment in adulthood. We also took a developmental approach and examined associations between the PGSs for smoking and educational attainment and several intermediate phenotypes that contribute to educational success including grades, academic motivation, and disciplinary problems in childhood and adolescence. We operationalized these intermediate academic phenotypes using the stable variance across multiple occasions (ages 11, 14, and 17-years old), which removed time-specific influences and unsystematic measurement error from these measures. We then tested whether the PGSs for educational attainment and smoking had incremental effects over and above each other, and if their effects differed across the different facets of academic adjustment in adolescence and educational attainment in adulthood (mean age 29.4 years). Associations between the PGSs and the intermediate academic variables can be conceptualized as examples of gene-environment correlation processes, wherein genetic dispositions influence the environments people shape or are exposed to, which then influences later outcomes [30, 31]. That is, some genetic influences are mediated through environmental experiences. Consequently, we also fit a path analysis model to delineate the effects of the PGSs for educational attainment and smoking on the intermediate academic variables in adolescence, and tested whether these intermediate variables mediated the associations of the PGSs on the more distal adulthood outcome of educational attainment. Finally, we also examined whether the expressed phenotype of cigarettes per day from ages 14 to 24 years old accounted for the association between the smoking PGS and educational attainment, or if the effect of the smoking PGS was mediated through the academic variables, which would be more consistent with a general effect of behavioral disinhibition.

## Methods

### Participants

Participants were members of the Minnesota Twin Family Study (MTFS), a longitudinal study of 3762 (52% female) twins (1881 pairs) [32]. All twin pairs were the same sex and lived with at least one biological parent within driving distance to the University of Minnesota laboratories when recruited. Exclusion criteria included any cognitive or physical disability that would interfere with study participation. Twins were recruited the year they turned either 11-years old ($n$ = 2510; 'younger cohort') or 17-years old ($n$ = 1252; 'older cohort'). Twins in the older cohort were born between 1972 and 1979, while twins in the younger cohort were born from 1977 to 1984 and 1988 to 1994. Families were representative of the recruitment area on

socioeconomic status, history of mental health treatment, and urban-rural residence [33]. Consistent with the demographics of Minnesota for the target birth years, 96% of participants reported non-Hispanic White ethnicity and race. All study protocols were evaluated and approved by the Institutional Review Board at the University of Minnesota. Written consent was obtained from all participants ages 18 years-old and older; written consent from parents and written assent from participants was obtained for all participants under age 18 years-old.

The younger cohort was assessed at ages 11 ($M_{age}$ = 11.78 years, SD = 0.43 years) and 14 ($M_{age}$ = 14.90 years, SD = 0.31 years), and all twins were assessed at ages 17 ($M_{age}$ = 17.85 years, SD = 0.64 years), 21 ($M_{age}$ = 21.08 years, SD = 0.79 years), and 24 ($M_{age}$ = 24.87 years, SD = 0.94 years). All twins from the 1972–1979 and 1977–1984 birth cohorts were also assessed at age 29 ($M_{age}$ = 29.43 years, SD = 0.67 years), and a subset (n = 866) of the latter cohort was also assessed at age 34 ($M_{age}$ = 34.62 years, SD = 1.30 years). Table 1 provides the number of participants and descriptive statistics for the measures of academic adjustment at ages 11, 14, and 17. Retention rates were 91.4% and 86.3% at ages 14 and 17, respectively, for the younger cohort. The total sample included 1205 monozygotic (51.5% female) and 676 dizygotic (52.8% female) twin pairs.

## Assessment

**Grade Point Average (GPA).** Twins and their mothers reported on the grades twins typically received in reading/English, math, social studies/history, and science classes by indicating whether the grades were much better than average (A = 4), above average (B = 3), average (C = 2), below average (D = 1) or very much below average, failing (F = 0). This approach was taken to standardize grade assessment and facilitate comparison since participants attended different school districts that employed different grading formats, procedures, and standards. We used the mean rating across class subjects for the GPA variable, and averaged the GPA scores across twin and mother reports (r = .79). The validity of this approach was tested using 67 school transcripts from a random sample of younger cohort twins, and the correlation between the MTFS rating and actual grades was r = .89 [9]. Participants who dropped out of high school reported grades for the last year they attended school.

**Academic motivation.** Twins and their mothers completed a 6-item (α = .83) scale assessing twins' attitudes about school (*interested in school work*; *enjoys attending school*; *turns in assignments on time*; *liked by teachers*; *has a good attitude about school*; *motivated to earn good grades*) [9]. We used the mean of the self and mother reports (r = .51) for the academic motivation score.

**Table 1. Descriptive information for academic adjustment variables at ages 11, 14, and 17.**

| | Age 11 | | | Age 14 | | | | Age 17 | | | | |
|---|---|---|---|---|---|---|---|---|---|---|---|---|
| | M | SD | N | M | SD | N | r11 | M | SD | N | r11 | r14 |
| Grade Point Average | 3.10 | .66 | 2492 | 3.06 | .79 | 2357 | .64 | 3.02 | .77 | 3449 | .56 | .73 |
| Academic Motivation Self/Parent Report | 20.89 | 2.50 | 2502 | 19.83 | 2.94 | 2335 | .47 | 19.76 | 3.08 | 3412 | .41 | .62 |
| Academic Motivation Teacher Report | 19.87 | 3.13 | 1754 | 19.64 | 3.40 | 1808 | .58 | 19.67 | 3.42 | 2249 | .53 | .58 |
| Disciplinary Problems | .00 | .61 | 2429 | .00 | .80 | 2357 | .25 | .00 | .75 | 3512 | .25 | .48 |

Note. M = mean; SD = standard deviation; N = number of ratings; r11 = correlation with corresponding variable at age 11; r14 = correlation with corresponding variable at age 14.

Teachers also rated twins on the same items using a teacher rating form that was completed by up to 3 teachers nominated by the twins. We used the mean rating across teachers whenever more than one teacher rating was available (~75% of participants with teacher rating data had at least two teacher informants). Teacher ratings were collected at each assessment and were available for 69.9%, 72.0%, and 59.8% of participants at ages 11, 14, and 17, respectively. It was Minnesota state policy to place twins from the same pair in separate classrooms whenever possible, which minimized bias due to twin contrast or comparison. The correlation between the teacher and self/mother ratings of academic motivation was $r = .55$.

**Disciplinary problems.** Twins and mothers reported on twins receiving school disciplinary actions for misbehavior including: sent to detention or held after school; sent to principal's office; notes sent home or parents called about student's behavior; parent-teacher conferences regarding student's behavior; skipping school or cutting classes; suspended or expelled from school. Responses were coded as 0 = *never*, 1 = *once or twice*, 2 = *two or more times*, and a behavior was considered present if reported by either the twin or mother ($r = .62$). We estimated disciplinary problems factor scores by fitting a 1-factor confirmatory factor analysis model to the six discipline problems items (mean factor loading = .83).

**Cigarettes per day.** Smoking was assessed using the average number of cigarettes smoked per day (or equivalent amount of an alternative form of nicotine use such as chews, cigars, etc.) at the target ages of 14, 17, 21, and 24 years old. Free responses were converted to a 0 (no use) to 6 (20 or more cigarettes per day) scale.

**Educational attainment.** We used the last assessment that a twin reported on their educational experiences (mean and median age 29.4 years, range 19.7 to 39.9 years) to code their highest level of educational attainment ($n = 3463$; 92.1% of the total sample). The educational attainment variable was coded as follows: 1 = less than high school diploma (9.5%), 2 = high school graduate or GED (9.8%), 3 = vocational degree, some college, or an associate's degree (30.5%), 4 = bachelor's degree (31.9%), 5 = master's level degree (8.8%), 6 = PhD or other advanced professional degree (e.g., MD, JD) (4.2%). Educational attainment was coded as missing (7.9%; $n = 299$) if the participant did not have data for a post high school assessment (i.e., age 20 or older). Because there was a significant correlation between age and educational attainment ($r = .27$, $p < .001$), we regressed educational attainment on age, and used the unstandardized residual score in all analyses.

**PGS methods.** We generated PGSs for smoking and educational attainment from the GWAS summary statistics of large discovery samples for having ever smoked regularly [25] and years of education [11], after removing the MTFS sample that contributed to those GWAS to remove overlap with our study sample. We created smoking PGSs for participants of European ancestry in the MTFS target sample following imputation to the most recent Haplotype Reference Consortium reference panel [34, 35], and restricted to variants with minor allele frequency $\geq .01$ and with imputation quality scores greater than 0.7. For the educational attainment PGS, we applied the same QC procedure for summary statistics of educational attainment GWAS [11] and additionally removed the MHC region (chr6:28477797–33448354). We then generated beta weights in the MTFS sample for the resulting ~1 million filtered HapMap3 variants using LDpred v.1.0 [36], including variants of all significance levels (i.e., $p \leq 1$) to capture all genetic influences across the genome. We then calculated smoking and educational attainment PGSs in PLINK 1.9 [37] for all participants meeting this study's inclusion criteria ($n = 3225$).

## Data analytic strategy

We examined associations among the smoking and educational attainment PGSs and the longitudinal measures of academic adjustment using multiple regression models and random intercept panel models (RI-PM; see Fig 1). In the multiple regression models, we entered an academic variable at a single time point as the outcome and regressed on the smoking PGS or the educational attainment PGS, as well as the covariates of participant sex and the first five genetic principal components to adjust for ancestral stratification [38].

We then fit univariate, unconditional RI-PMs to the longitudinal measures of academic adjustment ($GPA_{11}$, $GPA_{14}$, $GPA_{17}$ in Fig 1). In each model, we specified the measures of a given academic variable at each time point to load on a time-invariant random intercept factor (RI in Fig 1). We fixed factor loadings to 1, and allowed indicator intercepts to vary. We fixed the mean of the random intercept to 0 and estimated its variance freely. The random intercept captured the variance in the indicators shared across time points, that is, the stable trait variance across time [39]. For example, a positive random intercept score indicates that an individual consistently ranked higher than the sample mean across time points. We specified occasion-specific residual factors ($R_{11}$ through $R_{17}$ in Fig 1) with factor loadings fixed to 1, means fixed to 0, and variances freely estimated. We added autoregressive paths from one residual factor to the subsequent residual factor. These paths captured the extent to which time point-specific deviations at one time-point were related to time point-specific deviations at the subsequent time points, and were included because not accounting for residual autoregressive variance could lead to biased variance estimates in the intercept factors [40, 41]. We then fit conditional models in which the random intercept factors were either regressed on a single PGS and the control variables (1 PGS model), or both PGSs and the control variables (2 PGS model). We fit a similar conditional RI-PM for cigarettes per day, except that we used data for ages 14, 17, 21, and 24 years old, as there was very little smoking at age 11, and levels of cigarettes per day tend to peak in the early to mid-20's.

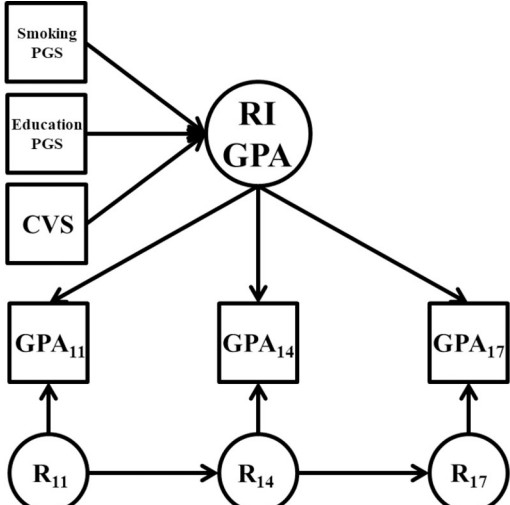

**Fig 1. Conditional random intercept model.** PGS = polygenic risk score; RI = random intercept factor; GPA = grade point average; R = residual factors at age 11, 14, and 17; CVS = set of covariates including the first five genetic principal components, sex. The two PGS Predictor model for GPA is specifically depicted here for illustrative purposes, but all conditional random intercept models followed this general structure (with either one or two PGS predictor variables). Variances/residual variances and mean structure omitted from figure for clarity of presentation.

Finally, we fit a path analysis mediation model to predict educational attainment in adulthood (Fig 2). In this model, the smoking PGS, educational attainment PGS, and control variables (sex and ancestry principal components) were the independent variables, and scores on the random intercepts of the adolescent academic variables and cigarettes per day were the mediator variables. To increase the model's computational feasibility, we first estimated factor scores for the five random intercepts (via maximum a posteriori estimation) to include in the path analysis model. We specified paths from the independent variables to the five random intercept scores and educational attainment, and from the five random intercepts to educational attainment. We included covariances between all independent variables and specified residual covariances among the random intercept variables.

We fit all models in Mplus version 8.4 [42] using full information maximum likelihood estimation. We derived confidence intervals using clustered (by family) nonparametric percentile bootstrap (1000 draws), which provides reliable assessments of parameter estimate precision under a variety of complex data conditions [43]. We considered a parameter estimate statistically significant if the bootstrapped 95% confidence interval did not include 0, and its *p*-value was < .005. We used the Mplus Automation Package [44] in R [45] to facilitate the analyses.

## Results

Descriptive information for the academic variables including the *N's* at ages 11, 14, and 17 and autocorrelations are reported in Table 1. The GPA (mean autocorrelation = .69), academic motivation (mean autocorrelation = .55), teacher rating of academic motivation (mean autocorrelation = .58), and discipline problems (mean autocorrelation = .37) measures had moderate to high stability over time. Mean-levels of cigarettes per day increased through the age 14 (M = 0.53, SD = 1.29, *n* = 2334), age 17 (M = 1.36, SD = 1.86, *n* = 3444), and age 21 (M = 2.00,

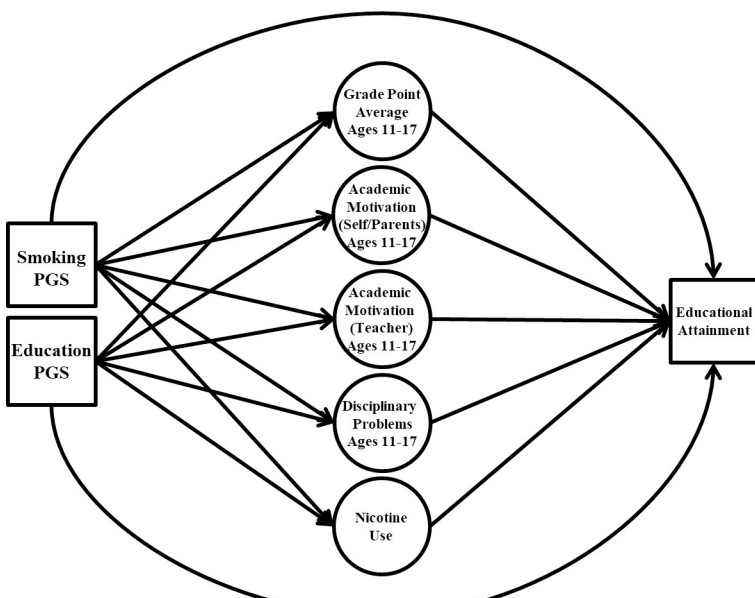

**Fig 2. Educational attainment mediation model.** PGS = polygenic risk score. Circles represent random intercept factor scores. The first five genetic principal components and sex were included alongside the PGSs as predictors of the random intercept factor scores and educational attainment in adulthood. Covariate paths, covariances between predictor variables, covariances between random intercepts, variances/residual variances, and mean structure omitted from figure for clarity of presentation. Random intercepts represented via random intercept factor scores from the initial, unconditional random intercept models.

SD = 2.06, *n* = 2698) assessments, and then declined slightly at the age 24 (M = 1.79, SD = 2.05, *n* = 3258) assessment, and exhibited moderate to high rank-order stability (mean autocorrelation = .67). The unconditional RI-PMs were fully saturated and thus perfectly fit the data. The variance component of each random intercept was statistically significant, and the residual structure autoregressive coefficients were small to moderate (mean coefficients of .21 from age 11 to 14, and .40 from age 14 to 17).

## GPA

Results for the multiple regression models and RI-PMs for GPA, academic motivation, teacher rating of academic motivation, and disciplinary problems are presented in Table 2. For GPA, all the regression coefficients were statistically significant and small to medium in size for both the smoking (mean $\beta$ = -.15) and educational attainment (mean $\beta$ = .23) PGS. In the RI-PMs, the associations for the smoking ($\beta$ = -.20, 95% CI: -.27, -.14) and educational attainment ($\beta$ = .35, 95% CI: .29, .42) PGSs were larger than those in the multiple regression models, as expected given removal of time-specific variance including measurement error. When the two PGSs were included in the same RI-PM, both the smoking ($\beta$ = -.13, 95% CI: -.19, -.07) and educational attainment ($\beta$ = .32, 95% CI: .26, .40) PGSs remained statistically significant, though the effect size for the smoking PGS decreased by about 35%.

### Academic motivation

All regression coefficients were statistically significant for the associations between the measures of academic motivation and the smoking (mean $\beta$'s = -.14 and -.16 for self/parent and

**Table 2. Standardized regression coefficients from smoking and education polygenic scores (PGS) to outcome variables.**

|  | Smoking PGS | | | | | Education PGS | | | | |
|---|---|---|---|---|---|---|---|---|---|---|
|  | **Age 11** | **Age 14** | **Age 17** | **RI** | **RI** | **Age 11** | **Age 14** | **Age 17** | **RI** | **RI** |
|  |  |  |  | **1 PGS** | **2 PGS** |  |  |  | **1 PGS** | **2 PGS** |
| Grade Point Average | **-.13** | **-18** | **-.15** | **-.20** | **-.13** | **.20** | **.25** | **.25** | **.35** | **.32** |
|  | [-.18, -.07] | [-.23, -.13] | [-.19, -.11] | [-.27, -.14] | [-.19, -.07] | [.15, .26] | [.21, .30] | [.20, .29] | [.29, .42] | [.26, .40] |
| Academic Motivation | **-.10** | **-.17** | **-.16** | **-.22** | **-.19** | **.06** | **.14** | **.15** | **.17** | **.12** |
| Self/Parent Report | [-.15, -.05] | [-.22, -.13] | [-.20, -.12] | [-.28, -.16] | [-.26, -.13] | [.01, .10] | [.10, .19] | [.10, .18] | [.11, .23] | [.07, .19] |
| Academic Motivation | **-.14** | **-.19** | **-.15** | **-.21** | **-.15** | **.19** | **.25** | **.18** | **.28** | **.25** |
| Teacher Report | [-.19, -.07] | [-.24, -.14] | [-.20, -.10] | [-.27, -.15] | [-.21, -.09] | [.13, .25] | [.20, .30] | [.13, .23] | [.23, .34] | [.19, .31] |
| Disciplinary Problems | **.08** | **.17** | **.16** | **.28** | **.26** | -.03 | **-.12** | **-.08** | **-.15** | **-.09** |
|  | [.03, .13] | [.12, .21] | [.12, .20] | [.20, .36] | [.18, .34] | [-.08, .02] | [-.17, -.07] | [-.04, -.12] | [-.22, -.07] | [-.16, -.01] |
| Cigarettes per day | -- | -- | -- | **.20** | **.18** | -- | -- | -- | **-.14** | **.10** |
|  |  |  |  | [.16, .24] | [.14, .21] |  |  |  | [-.18, -.10] | [-.15, -.06] |
| Educational Attainment | -- | -- | -- | **-.19** | **-.14** | -- | -- | -- | **.26** | **.23** |
|  |  |  |  | [-.23, -.15] | [-.18, -.10] |  |  |  | [.22, .30] | [.19, .27] |

Note. Age 11 = regression paths from PGS to outcome variable at age 11; Age 14 = regression paths from PGS to outcome variable at age 14; Age 17 = regression paths from PGS to outcome variable at age 17; RI = random intercept factor; 1 PGS = coefficients from one PGS predictor model; 2 PGS = coefficients from two PGS predictor model; Bold = 95% confidence interval does not include 0. Only one PGS was entered as a predictor in each one PGS predictor model; both PGSs were entered as predictors simultaneously in the two PGS predictor model. Smoking is a random intercept factor score for nicotine quantity assessed at ages 14, 17, 21, and 24 years old. Educational attainment was adjusted for age (mean and median age 29.4 years, SD = 3.9 years, range 19.7 to 39.9 years). All models include participant sex and first five genetic principal components as covariates (covariate regression paths not included). 95% confidence intervals presented under estimates; estimates for which confidence intervals do not include 0 are presented in bold. Confidence intervals derived via clustered, non-parametric percentile bootstrapping (with 1,000 random draws).

teacher ratings, respectively) and educational attainment (mean $\beta$'s = .12 and .21, for self/parent and teacher ratings, respectively) PGS. In the RI-PM, associations with the random intercept factors for the self/parent and teacher ratings of academic motivation were slightly larger for both the smoking (mean $\beta$ = -.22) and educational attainment (mean $\beta$ = .23) PGSs. When the two PGSs were included in the same RI-PMs, the adjusted associations with the smoking (mean $\beta$ = -.17) and educational attainment (mean $\beta$ = .19) PGSs remained statistically significant, with an average reduction in effect sizes of about 20%.

## Disciplinary problems

All regression coefficients were statistically significant for the associations between disciplinary problems and the smoking PGS (mean $\beta$ = .14). Associations between disciplinary problems and the educational attainment PGS were significant at ages 14 ($\beta$ = -.12, 95% CI: -.17, -.07) and 17 ($\beta$ = -.08, 95% CI: -.12, -.04, respectively), but not age 11 ($\beta$ = -.03, 95% CI: -.08, .02). In the RI-PM, associations with the random intercept factor of disciplinary problems was much larger for the smoking PGS ($\beta$ = .28, 95% CI: .20, .36) and slightly larger for the educational attainment PGS ($\beta$ = -.15, 95% CI: -.22, -.07). When the two PGSs were included in the same RI-PM, the adjusted associations between the random intercept factor and the smoking ($\beta$ = .26, 95% CI: .18, .34) and educational attainment ($\beta$ = -.09, 95% CI: -.16, -.01) PGSs remained statistically significant, though the effect size for the educational attainment PGS decreased by about 40%.

## Cigarettes per day

In the RI-PM, both the smoking ($\beta$ = .20, 95% CI: .16, .24) and educational attainment ($\beta$ = -.14, 95% CI: -.18, -.10) PGSs had significant associations with the random intercept factor for cigarettes per day (see Table 2). These effects remained significant after adjusting for their overlap, though the effects declined by about 29% for the educational attainment PGS and 10% for the smoking PGS.

## Education attainment in adulthood

Both the smoking ($\beta$ = -.19, 95% CI: -.24, -.15) and educational attainment ($\beta$ = .26, 95% CI: .22, .30) PGSs had significant associations with educational attainment in adulthood (see Table 2). These effects remained significant after adjusting for their overlap, though the effects declined by about 26% for the smoking PGS and 12% for the educational attainment PGS. Table 3 includes the correlations among the smoking and educational PGSs, estimated scores for random intercept factors of the four academic variables in adolescence and cigarettes per day, and educational attainment in adulthood. The four academic variables had large associations with each other (mean $r$ = |.53|) and educational attainment ($r$'s = |.35| to |.52|; $R^2$ = .34). Cigarettes per day also had a robust association with educational attainment ($r$ = -.30).

Results from the mediation model that estimated the direct and indirect effects of the smoking and educational attainment PGSs via the academic variables in adolescence and cigarettes per day on educational attainment in adulthood are presented in Table 4. Inclusion of the smoking and educational attainment PGSs resulted in a significant increase in $\Delta R^2$ = .02 ($R^2$ = .36; $\Delta\chi^2(2)$ = 89.56, $p < .001$) over and above the four adolescent academic variables and cigarettes per day. Both the smoking ($\beta$ = |.09| to |.18|) and educational attainment ($\beta$ = |.05| to |.22|) PGSs had significant associations on the random intercept scores for each academic variable in adolescence and cigarettes per day.

Random intercept scores for GPA, teacher ratings of academic motivation, disciplinary problems, and cigarettes per day were in turn significantly associated with educational attainment in adulthood (last row Table 4). These effects adjusted for the common variance among

**Table 3. Correlations among variables in the mediation model.**

|  | 1 | 2 | 3 | 4 | 5 | 6 | 7 | 8 |
|---|---|---|---|---|---|---|---|---|
| 1. Smoking PGS |  |  |  |  |  |  |  |  |
| 2. Education PGS | -.23 |  |  |  |  |  |  |  |
| 3. RI-GPA | -.14 | .24 |  |  |  |  |  |  |
| 4. RI-Academic Motivation Self/Parent report | -.15 | .11 | .58 |  |  |  |  |  |
| 5. RI-Academic Motivation Teacher report | -.17 | .21 | .67 | .59 |  |  |  |  |
| 6. RI-Disciplinary Problems | .16 | -.08 | -.38 | -.47 | -.51 |  |  |  |
| 7. RI-Cigarettes per day | .20 | -.14 | -.30 | -.36 | -.38 | .37 |  |  |
| 8. Educational Attainment | -.20 | .26 | .52 | .39 | .51 | -.35 | -.30 |  |

PGS = polygenic score; RI = random intercept; GPA = grade point average. Genetic principal components 1 through 5 and sex were also included in the mediation model as control variables; correlations for these control variables not presented. Random intercept correlations based on factor scores derived from the initial unconditional random intercept models. Educational attainment was adjusted for age (mean and median age 29.4 years, SD = 3.9 years, range 19.7 to 39.9 years).

**Table 4. Standardized coefficients from educational attainment mediation model.**

|  | Random Intercept Factors | | | | | Educational Attainment |
|---|---|---|---|---|---|---|
|  | GPA | Academic Motivation (Self/Parent) | Academic Motivation (Teacher) | Disciplinary Problems | Cigarettes per day | |
| Smoking PGS |  |  |  |  |  |  |
| PGS → RI | **-.09** | **-.13** | **-.13** | **.15** | **.18** | -- |
|  | **[-.13, -.05]** | **[-.17, -.09]** | **[-.17, -.09]** | **[.11, .19]** | **[.14, .22]** |  |
| PGS → Educational Attainment | -- | -- | -- | -- | -- | **-.07** |
|  |  |  |  |  |  | **[-.11, -.04]** |
| PGS → RI → Educational Attainment | **-.03** | .00 | **-.03** | **-.01** | -.01 | **-.08** |
|  | **[-.04, -.01]** | [-.01, .00] | **[-.04, -.02]** | **[-.02, -.01]** | [-.02, .00] | **[-.10, -.05]** |
| Education PGS |  |  |  |  |  |  |
| PGS → RI | **.22** | **.09** | **.19** | **-.05** | **-.10** | -- |
|  | **[.18, .27]** | **[.05, .13]** | **[.15, .23]** | **[-.10, -.01]** | **[-.14, -.06]** |  |
| PGS → Educational Attainment | -- | -- | -- | -- | -- | **.11** |
|  |  |  |  |  |  | **[.08, .15]** |
| PGS → RI → Educational Attainment | **.06** | .00 | **.04** | .00 | -.01 | **.12** |
|  | **[.05, .08]** | [.00, .01] | **[.03, .05]** | [.00, .01] | [.00, .01] | **[.10, .14]** |
| Educational Attainment |  |  |  |  |  |  |
| RI → Educational Attainment | **.28** | .03 | **.21** | **-.08** | **-.05** | -- |
|  | **[.24, .33]** | [-.02, .07] | **[.16, .26]** | **[-.12, -.04]** | **[-.09, -.02]** |  |

Note. PGS = polygenic score; RI = random intercept; GPA = grade point average; PGS → RI = paths from the PGS to the random intercept; PGS → Educational Attainment = paths from the PGS to educational attainment (i.e., the direct effect); PGS → RI → Educational Attainment = indirect effects from the PGS to educational attainment through the corresponding RI (final entry in these rows correspond to the total indirect effect through all RIs); RI → Educational Attainment = paths from the RI to educational attainment; Bold = 95% confidence interval does not include 0. All paths come from a single model with one outcome variable (educational attainment), five intervening variables (the random intercepts), and 6 independent variables (the PGSs and covariates of sex and 5 genetic principal components to adjust for ancestry; coefficients for control variables not presented). In this model, the random intercepts were represented via random intercept factor scores from the initial unconditional random intercept models. Confidence intervals derived via clustered non-parametric percentile bootstrap with 1,000 draws. Educational attainment was adjusted for age (mean age 29.4 years, SD = 3.9 years, range 19.7 to 39.9 years). $R^2$ = .36.

all the predictors, and so were substantially smaller than the unadjusted correlations, but were still robust for GPA ($\beta$ = .28, 95% CI: .24, .28) and teacher ratings of academic motivation ($\beta$ = .21, 95% CI: .16, .26) and small for disciplinary problems ($\beta$ = -.08, 95% CI: -.12, -.04) and cigarettes per day ($\beta$ = -.05, 95% CI: -.09, -.02). Consequently, the smoking and educational attainment PGSs each had small but statistically significant indirect effects on educational attainment via GPA and teacher ratings of academic motivation in adolescence, and the smoking PGS also had a small indirect effect via disciplinary problems.

Cumulatively, the random intercept scores for the four academic variables and cigarettes per day accounted for about 50% of the adjusted effects of the smoking ($\beta$ = -.08, 95% CI: -.10, -.05) and educational attainment ($\beta$ = .12, 95% CI: .10, .14) PGSs on educational attainment in adulthood. Finally, the smoking ($\beta$ = -.07, 95% CI: -.11, -.04) and educational attainment ($\beta$ = .12, 95% CI: .10, .14) PGSs continued to have small but significant direct effects on educational attainment in adulthood, even after adjusting for their overlap, the four adolescent academic variables and cigarettes per day.

## Discussion

The results provided strong evidence that PGSs for smoking and for educational attainment each predicted educational attainment in adulthood. Most importantly, our analyses demonstrated that genetic influences on smoking provide incremental prediction of educational attainment, even after accounting for a PGS specifically designed to predict educational attainment. This indicates that PGSs calibrated on different phenotypes can provide additional information about genetic influences on a target phenotype, even ones that have already been the subject of large gene discovery analyses. Our results also illustrate the complexity of a distal outcome such as educational attainment, which is the result of the cumulative influences of numerous genetic and environmental processes.

To begin to parse these processes, we took a developmental approach and also examined associations between the smoking and educational attainment PGSs and several variables associated with academic adjustment in childhood and adolescence. Interestingly, both the smoking and educational attainment PGSs had at least small and significant associations with each facet of academic adjustment we examined, indicating each PGS measures non-specific genetic influences that contribute to a variety of intermediate academic variables. However, there were some indications of specificity for the PGSs, especially after adjusting for their overlap. Specifically, the educational attainment PGS had its strongest association with GPA while the strongest association with the smoking PGS was with disciplinary problems, and the strongest association for one PGS was the weakest for the other (see Table 4). These results are relatively intuitive given that grades in middle and high school were the most predictive of the variables of later educational attainment that we examined [5], and disciplinary problems were the variables most strongly associated with externalizing problems of which smoking is highly correlated [20].

Given the non-specific associations of both the smoking and educational attainment PGSs, it will be important to continue to establish their construct validity, that is, what these scores measure in terms of their phenotypic associations and the biological processes associated with the specific genes driving their effects. Substantial evidence is mounting that the smoking PGS measures the broader construct of behavioral disinhibition rather than the narrow phenotype of nicotine addiction [19], given its associations with the use of multiple substance classes, externalizing problems, antisocial peers, facets of poor academic adjustment, and low educational attainment [25–29]. The failure to detect an indirect effect of the smoking PGS on educational attainment via cigarettes per day is further evidence that the smoking PGS taps a

broader behavioral style than risk for nicotine addiction specifically. The educational attainment PGS has now been associated with longitudinal measurements of reading skills, mental age of IQ tests, and grades, suggesting it measures processes associated with cognitive development and academic skill acquisition [46]. Both PGSs, however, seem to index processes that are eventually expressed as broad psychological processes that likely have cognitive (e.g., intellectual abilities), affective (e.g., positive emotions related to academic activities), and behavioral (e.g., ability to follow directions, persist in tasks, withhold responses) components that contribute to numerous traits and life outcomes [6]. Though GWAS and PGSs are important advances in behavioral genetic methods, delineating the numerous biological, contextual, and psychological linkages between specific genetic markers and life outcomes such as educational attainment will continue to be a complex task.

Cumulatively, the intermediate academic variables accounted for about 50% of the adjusted effects of the PGSs on educational attainment in adulthood. Defining the 'environment' broadly, these academic variables are proxies for some aspects of educational context. Because they are genetically influenced, their role in educational attainment reflects gene-environment correlation processes wherein genetic influences contribute to exposure to experiences that then contribute to later educational attainment [30, 31]. For example, genetic influences that contribute to better grades and greater academic motivation likely contribute to receiving more rewards, reinforcement, encouragement from adults for pursuing academic activities, and admission to higher education, which further facilitates reaching the maximal phenotypic expression of a person's genetic potential.

Notably, we were only able to account for about one third of the variability in educational attainment. This was in spite of several design strengths including inclusion of relevant academic variables assessed on multiple occasions using multiple informants and several facets of academic adjustment in addition to the two PGSs. Also, the sample was not racially or ethnically diverse, which reduces variability in the United States. Unassessed variables may account for substantial portions of additional variance in educational attainment, such as family attitudes about education and the availability of resources to contribute to obtaining higher levels of education [47]. Whether a person pursues advanced education, however, depends on both idiosyncratic and social-structure factors such as availability of job opportunities not requiring additional education, family and partner relationships, specific academic experiences (e.g., satisfying versus dissatisfying), financial constraints, stereotypes about pursuing certain fields of interest, and incentives to return to school after an extended hiatus. Such factors were not well captured in our models.

The study had other limitations. The PGSs did not identify specific genetic variants that point to biological processes that might account for their associations with educational attainment. Functional genomic information is needed to understand the biological processes accounting for these associations [48, 49]. Also, while the hope is that PGSs will eventually have practical value in predicting individual outcomes and informing intervention efforts, this is not yet viable given the small effect sizes. Further, the sample was restricted to people of European ancestry and persons growing up in Minnesota so it is unclear whether the results generalize to other ancestral groups with different allelic frequencies, or societies with different educational systems (e.g., societies with weaker educational infrastructure and fewer opportunities or those with universal access to higher education). Additionally, societal influences related to racial, ethnic, and gender inequities and discrimination in education and cultural values and resources committed to education might moderate genetic influences measured by the PGSs [50]. Given substantial overlap between ancestry status and socially defined racial/ethnic status, efforts to improve educational outcomes using PGS approaches have the potential to increase existing disparities if these findings are only applicable to people of European

ancestry or culturally defined White people, further prioritizing extending these kinds of studies to diverse ancestry and racial/ethnic groups [51].

Despite these limitations, this study extended prior work by demonstrating the incremental value of multiple PGSs to predict educational attainment and some of the intermediate phenotypes related to this distal outcome. Hopefully, continued validation of PGSs and delineation of linkages between biological and environmental processes will contribute to improved educational outcomes and human flourishing.

## Author Contributions

**Conceptualization:** Brian M. Hicks, Wendy Johnson.

**Formal analysis:** D. Angus Clark, Mengzhen Liu, Seonkyeong Jang.

**Funding acquisition:** William G. Iacono, Matt McGue, Scott I. Vrieze.

**Project administration:** William G. Iacono, Matt McGue, Scott I. Vrieze.

**Supervision:** Brian M. Hicks.

**Writing – original draft:** Brian M. Hicks, D. Angus Clark, Mengzhen Liu, Seonkyeong Jang.

**Writing – review & editing:** Brian M. Hicks, Joseph D. Deak, Jonathan D. Schaefer, C. Emily Durbin, Wendy Johnson, Sylia Wilson, William G. Iacono, Matt McGue, Scott I. Vrieze.

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
