## [Decision Letter · Decision Letter 0]

23 Apr 2021

PONE-D-21-06130

Polygenic scores for smoking and educational attainment have Independent Influences on academic success and adjustment in adolescence and educational attainment in adulthood

PLOS ONE

Dear Dr. Hicks,

Thank you for submitting your manuscript to PLOS ONE. After careful consideration, we feel that it has merit but does not fully meet PLOS ONE’s publication criteria as it currently stands. Therefore, we invite you to submit a revised version of the manuscript that addresses the points raised during the review process.

I was able to obtain two reviews from scholars with expertise on the topic and methods of your study. I reviewed the paper independently. The reviewers were positive about the article; it is well written, with a large and informative dataset. The paper is clear in focus, and the analytic models are thoroughly described and justified. There are some issues that would need to be resolved if the paper is to be accepted for publication in PLOS ONE. First, both reviewers requested more justification for the use of the nicotine PGS to capture genetic variance associated with behavioral disinhibition, which should relate to eventual educational attainment. I was also curious about how you would interpret the positive correlation between the two PGS scores (.23), and their opposite relationships with external criteria. Second, more information is needed on the retention rate by age 29, and how many participants were missing educational attainment data at age 29 (and what assessment time point was used as a substitution in those cases). Third, the size and meaningfulness of the effect sizes can be emphasized much more, and conclusions should align with expected impact of the findings.

As Reviewer 2 also noted, the correlations between the education PGS and academic variables are shown as negative in Table 3, whereas the regression coefficients (Table 2) are all positive. I assume these are typo’s?

The reviewers made other comments that you should attend to in your response and revision. Thank you for submitting your research for consideration in PLOS ONE.

We look forward to receiving your revised manuscript.

Kind regards,

Edelyn Verona

Academic Editor

PLOS ONE

Journal Requirements:

PLOS requires an ORCID iD for the corresponding author in Editorial Manager on papers submitted after December 6th, 2016. Please ensure that you have an ORCID iD and that it is validated in Editorial Manager. To do this, go to ‘Update my Information’ (in the upper left-hand corner of the main menu), and click on the Fetch/Validate link next to the ORCID field. This will take you to the ORCID site and allow you to create a new iD or authenticate a pre-existing iD in Editorial Manager. Please see the following video for instructions on linking an ORCID iD to your Editorial Manager account: https://www.youtube.com/watch?v=_xcclfuvtxQ

In your Data Availability statement, you have not specified where the minimal data set underlying the results described in your manuscript can be found. PLOS defines a study's minimal data set as the underlying data used to reach the conclusions drawn in the manuscript and any additional data required to replicate the reported study findings in their entirety. All PLOS journals require that the minimal data set be made fully available. For more information about our data policy, please see http://journals.plos.org/plosone/s/data-availability.

Reviewers' comments:

Reviewer's Responses to Questions

**Comments to the Author**

1. Is the manuscript technically sound, and do the data support the conclusions?

Reviewer #1: Yes

Reviewer #2: Yes

2. Has the statistical analysis been performed appropriately and rigorously? 

Reviewer #1: Yes

Reviewer #2: Yes

3. Have the authors made all data underlying the findings in their manuscript fully available?

Reviewer #1: No

Reviewer #2: No

4. Is the manuscript presented in an intelligible fashion and written in standard English?

Reviewer #1: Yes

Reviewer #2: Yes

5. Review Comments to the Author

Reviewer #1: This is a nice and clear paper on the relationship of PRS for nicotine and educational attainment on academic process indicators (motivation, gpa, disciplinary issues) as well as measured educational achievement at age 29. A mediation analysis showed that academic process indicators partially mediated relationship between PRSs and outcome. I have several concerns, questions, and suggestions for this paper.

First, it might be good to be clear why PRS for nicotine and educational attainment – rather than for behavioral disinhibition and cognitive ability – are being used. I understand the argument that the nic PRS predicts a host of externalizing psychopathology, but one could make the argument that it’s missing some of the effects to broader externalizing/behavioral disinhibition.

In the literature, are there any LD regressions that document the rG between educational attainment and nicotine?

I was not entirely clear how the authors are testing rGE from their into setup or their methods. Academic process and outcome variables are going to be influenced by both genes and environment, most certainly. How is a correlation between e.g., nicotine PRS with academic process variables a test of rGE? Either be clear about the logic or tone it down.

Educational attainment was taken from age 17 assessment if age 29 assessment was missing. If someone has e.g., really good grades and is academically on track at age 17, but fails to come in at age 29 is classified as ‘completed high school’, isn’t that a possible misclassification? How do the authors get around that and why not just treat the missing people as missing? Also, what were the retention rates at age 29 from baseline and also from age 17 – maybe I’m missing this portion?

Table 2 was a tad confusing, especially the columns relating to the 2 PRS analysis. If there are two PRSs in that analysis, why is there only a single value for each criterion? I might be missing something obvious here, perhaps.

Possible rater and sex effects. The authors are using sex as a covariate, and they are also collapsing mom and child reports of educational process. This is reasonable, but the readers might also want to know if there are rater and sex effects. It might be good to redo the models by rater and separately by sex and supplement the information (even though the sex effects models might have power issues, they might also highlight if the effects are particularly strong in one sex).

Typo – page 13 – last line in bracket – should be beta rather than a square (something happened with formatting)

Reviewer #2: # PONE-D-21-06130 Polygenic scores for smoking and educational attainment have Independent Influences on academic success and adjustment in adolescence and educational attainment in adulthood

In participants of the Minnesota Twin Family Study, polygenic scores (PGS) for smoking and educational attainment from large GWAMA consortia (after excluding sample overlap) proved to be significant and independent predictors of grades, academic motivation, and discipline problems at ages 11, 14, and 17 years-old, and of educational attainment at age 29. About half of the adjusted effects of the smoking and educational PGSs on educational attainment at age 29 were mediated by the academic variables in adolescence.

The paper is well-written and well-structured and the polygenic risk score methods used are sound and clearly described. My two concerns are (1) the use of a smoking PRS as a stand-in for a behavioural inhibition PRS and (2) the very small added predictive value of both PRS for educational attainment compared to the observed academic variables (3%).

The empirical results are based on a smoking PRS. I believe it would be best to stick with that concept up till the discussion. While I find the idea that the smoking PRS partly captures behavioral disinhibition not unreasonable, I see it more as an annotation of the results that should be put in the discussion and not to be claimed up front as early as the abstract and introduction as ‘a measure of genetic influences on behavioral disinhibition’.

The predictive effect sizes of the two PRS are very small – I find these results not very compatible with the strong conclusions in the summary in the discussion ‘The results provided strong evidence that PGSs for smoking—a measure of genetic influences on behavioral disinhibition—and for educational attainment each predicted educational attainment in adulthood. Most importantly, our analyses demonstrated that genetic influences on behavioral disinhibition provide incremental prediction of educational attainment, even after accounting for a PGS specifically designed to predict educational attainment.’

MINOR:

In table 3 the educational attainment PGS is negatively associated with the academic skills and with educational attainment at age 29. Is this an error or am I missing a reverse coding step? In table 2 the sign of the prediction beta seems OK. Confusing.

Does the .23 correlation between the PGS for smoking and educational attainment reflect pleiotropy? How does it affect the method used?

6. PLOS authors have the option to publish the peer review history of their article (what does this mean?). If published, this will include your full peer review and any attached files.

Reviewer #1: No

Reviewer #2: No

---

## [Author Response · Author response to Decision Letter 0]

1 Jun 2021

28 May 2021 

Dear Dr. Verona

We are submitting a revised manuscript entitled, “Polygenic Scores for Smoking and Educational Attainment have Independent Influences on Academic Success and Adjustment in Adolescence and Educational Attainment in Adulthood,” for publication in PLOS One. 

The review was very positive, but also noted several points for improvement including some errors in the original manuscript. Also, note that the data has now been uploaded to an OSF page (https://osf.io/92esr/).

Below, we describe revisions in response to feedback from you and the anonymous reviewers: 

1) Confusion regarding the direction of associations among variables:

“I was also curious about how you would interpret the positive correlation between the two PGS scores (.23), and their opposite relationships with external criteria.”

“As Reviewer 2 also noted, the correlations between the education PGS and academic variables are shown as negative in Table 3, whereas the regression coefficients (Table 2) are all positive. I assume these are typo’s?”

Unfortunately, there were several typos in the manuscript, which seems to have been a function having different analysts calculating each polygenic score (PGS) and a third lead analyst fitting the regression/SEM models. After a thorough review, all discrepancies have been resolved. Specifically, the smoking and educational attainment PGS are negatively correlated (r = -.23). The smoking PGS is negatively correlated with the educational attainment outcome and the adolescent academic variables, and positively correlated with disciplinary problems. The educational attainment PGS is positively correlated with the educational attainment outcome and the adolescent academic variables, and negatively correlated with disciplinary problems. 

2) More information about the educational attainment outcome:

“Second, more information is needed on the retention rate by age 29, and how many participants were missing educational attainment data at age 29 (and what assessment time point was used as a substitution in those cases).”

“Educational attainment was taken from age 17 assessment if age 29 assessment was missing. If someone has e.g., really good grades and is academically on track at age 17, but fails to come in at age 29 is classified as ‘completed high school’, isn’t that a possible misclassification? How do the authors get around that and why not just treat the missing people as missing? Also, what were the retention rates at age 29 from baseline and also from age 17 – maybe I’m missing this portion?”

We have since learned that the educational attainment variable is more complex than originally described. Rather than all participants reporting educational attainment at age 29, the educational attainment variable was coded based on the last post high school assessment the participant reported on their educational experiences. Consequently, while the mean, median, and modal age for educational attainment was 29.4 years old (SD = 3.9 years), the range was 19.7 to 39.9 years old (the older ages are due to a subset of twins participating in a target age 34 assessment). If a participant did not have information on educational experiences in adulthood (i.e., target age 20 assessment or older), their educational attainment was coded as missing. As such, educational attainment data was available for 95.2% (3070/3225) of the sample with PGS data and 92.1% (3463/3762) of the total sample. Because there was a significant correlation between age and educational attainment (r = .27, p < .001), we regressed educational attainment on age, and used the unstandardized residual score in all analyses. These details regarding the educational attainment outcome variable have been included in the Methods (pp. 10) and are provided below:

Educational Attainment. We used the last assessment that a twin reported on their educational experiences (mean and median age 29.4 years, range 19.7 to 39.9 years) to code their highest level of educational attainment (n = 3463; 92.1% of the total sample). The educational attainment variable was coded as follows: 1 = less than high school diploma (9.5%), 2 = high school graduate or GED (9.8%), 3 = vocational degree, some college, or an associate’s degree (30.5%), 4 = bachelor’s degree (31.9%), 5 = master’s level degree (8.8%), 6 = PhD or other advanced professional degree (e.g., MD, JD) (4.2%). Educational attainment was coded as missing (7.9%; n = 299) if the participant did not have data for a post high school assessment (i.e., age 20 or older). Because there was a significant correlation between age and educational attainment (r = .27, p < .001), we regressed educational attainment on age, and used the unstandardized residual score in all analyses. 

3) Conceptualization of smoking PGS as a measure of genetic influences for behavioral disinhibition: 

“First, both reviewers requested more justification for the use of the nicotine PGS to capture genetic variance associated with behavioral disinhibition, which should relate to eventual educational attainment.”

“First, it might be good to be clear why PRS for nicotine and educational attainment – rather than for behavioral disinhibition and cognitive ability – are being used. I understand the argument that the nic PRS predicts a host of externalizing psychopathology, but one could make the argument that it’s missing some of the effects to broader externalizing/behavioral disinhibition. In the literature, are there any LD regressions that document the rG between educational attainment and nicotine?”

“The empirical results are based on a smoking PRS. I believe it would be best to stick with that concept up till the discussion. While I find the idea that the smoking PRS partly captures behavioral disinhibition not unreasonable, I see it more as an annotation of the results that should be put in the discussion and not to be claimed up front as early as the abstract and introduction as ‘a measure of genetic influences on behavioral disinhibition’.”

We have made a number of changes to address these comments. First, we have added a paragraph describing the association between smoking and educational attainment, and the potential role for common genetic influences accounting for this association (pp. 5), see below: 

Smoking is a non-cognitive trait that has a strong association with lower educational attainment (13, 14). Rather than a direct causal effect of education on smoking (or vice versa), however, there has been a long recognition that this association is due to the common influences of third variables. For example, smoking patterns tend to be established in the late teens to early 20’s, which is prior to the completion of higher education but later than when consistent individual differences in factors strongly related to educational attainment (e.g., GPA, academic motivation, discipline problems) have emerged (15). Further, sib-pair difference analyses have found that familial factors account for the association between smoking and educational attainment (16). Finally, recent GWAS findings have estimated genetic correlations from r = .27 to .56 between smoking and educational attainment phenotypes, indicating at least some of their familial association is due to common genetic influences (17, 18). 

While we cut some references that substitutes the smoking PGS as a measure of behavioral disinhibition, we think it is important to retain the conceptualization that the smoking PGS operates more as a measure of genetic influences related to a broader behavioral disinhibition trait rather than a narrow nicotine addiction susceptibility. However, we now provide more rationale that smoking is highly correlated with externalizing problems and other substance use, which are all manifestations of a broad behavioral disinhibition trait. Further, we now frame the paper as a test of a hypothesis that the smoking PGS is associated with educational attainment due to its overlap with behavioral disinhibition. We also provide more details about recent studies consistent with hypothesis that the smoking PGS is a measure of genetic influences on behavioral disinhibition. See relevant text below, which directly follows the paragraph above (pp. 5-6)

Behavioral disinhibition refers to difficulty inhibiting impulses to behave in socially undesirable or restricted actions (19) and is a another non-cognitive trait that has been associated with academic success (4). Externalizing problems are manifestations of these poor inhibitory abilities and include impulsivity, aggression, rule breaking, oppositionality, hyperactivity, and inattention. They are associated with lower grades, poor academic motivation, and more disciplinary problems, and predict lower educational attainment (20, 21), with most of the overlap attributable to shared genetic influences (9). Smoking, especially in adolescence, is strongly correlated with externalizing behaviors, alcohol use, and other drug use, all of which are manifestations of a higher-order behavioral disinhibition trait (19, 22-24). It is possible then that the association between smoking and education attainment is actually due to the overlap between smoking and the broader trait of behavioral disinhibition. 

Recently, we examined the predictive validity of a PGS for having ever been a regular smoker that was derived from the largest GWAS of smoking-related phenotypes to date (N = 1,232,091)(25). In replication samples, this PGS accounted for 4% of the variance in a similar smoking phenotype and was also significantly associated with use measures of alcohol, cannabis, cocaine, amphetamines, ecstasy, and hallucinogens (25, 26). Using the same twin sample as in this report, we found that this smoking PGS predicted trajectories of nicotine and alcohol use from ages 14 to 34, even after adjusting for nicotine and alcohol use and a PGS for drinks per week (27). 

This smoking PGS was also associated with the externalizing dimension of the Child Behavior Checklist in a large sample of pre-adolescents, even after adjusting for a general factor of psychopathology (28). We followed up these results and found that the smoking PGS was associated with externalizing problems and personality traits associated with behavioral control—but not internalizing problems and extraversion—from ages 11 to 17 (29). We concluded that the smoking PGS was also a measure of genetic influences on general behavioral disinhibition rather than smoking or nicotine addiction specifically, and so could be used to investigate the role that genetic influences related to behavioral disinhibition have on the development of other near-neighbor outcomes. 

Finally, we added a random intercept variable for cigarettes per day at ages 14, 17, 21, and 24 years old to the mediation model. The rationale for this is to test well cigarettes per day, an expressed phenotype for the smoking PGS, either accounted for or provided an indirect path from the smoking PGS to educational attainment. We found that cigarettes per day did not account for the effect of the smoking PGS on educational attainment, and, in fact, the smoking PGS continued to have a significant direct effect on educational attainment even after accounting for all the other variables in the model. Further, cigarettes per day did not even provide a significant indirect path for the effect of the smoking PGS on educational attainment, but the academic variables of GPA and disciplinary problems did. We think this is additional and strong evidence that the smoking PGS measures genetic influences on a broad behavioral style consistent with conceptualizations of behavioral disinhibition. See text below from the Results (pp. 17-19): 

Cigarettes per day

In the RI-PM, both the smoking (standardized B = .20, 95% CI: .16, .24) and educational attainment (B = -.14, 95% CI: -.18, -.10) PGSs had significant associations with the random intercept factor for cigarettes per day (see Table 2). These effects remained significant after adjusting for their overlap, though the effects declined by about 29% for the educational attainment PGS and 10% for the smoking PGS. 

Education Attainment in adulthood

Both the smoking (standardized B = -.19, 95% CI: -.24, -.15) and educational attainment (B = .26, 95% CI: .22, .30) PGSs had significant associations with educational attainment in adulthood (see Table 2). These effects remained significant after adjusting for their overlap, though the effects declined by about 26% for the smoking PGS and 12% for the educational attainment PGS. Table 3 includes the correlations among the smoking and educational PGSs, estimated scores for random intercept factors of the four academic variables in adolescence and cigarettes per day, and educational attainment in adulthood. The four academic variables had large associations with each other (mean r = |.53|) and educational attainment (r’s = |.35| to |.52|; R2 = .34). Cigarettes per day also had a robust association with educational attainment (r = -.30). 

Results from the mediation model that estimated the direct and indirect effects of the smoking and educational attainment PGSs via the academic variables in adolescence and cigarettes per day on educational attainment in adulthood are presented in Table 4. Inclusion of the smoking and educational attainment PGSs resulted in a significant increase in 𝚫R2 = .02 (R2 = .36; 𝚫χ2(2) = 89.56, p < .001) over and above the four adolescent academic variables and cigarettes per day. Both the smoking (B = |.09| to |.18|) and educational attainment (B = |.05| to |.22|) PGSs had significant associations on the random intercept scores for each academic variable in adolescence and cigarettes per day. 

Random intercept scores for GPA, teacher ratings of academic motivation, disciplinary problems, and cigarettes per day were in turn significantly associated with educational attainment in adulthood (last row Table 4). These effects adjusted for the common variance among all the predictors, and so were substantially smaller than the unadjusted correlations, but were still robust for GPA (B = .28, 95% CI: .24, .28) and teacher ratings of academic motivation (B = .21, 95% CI: .16, .26) and small for disciplinary problems (B = -.08, 95% CI: -.12, -.04) and cigarettes per day (B = -.05, 95% CI: -.09, -.02). Consequently, the smoking and educational attainment PGSs each had small but statistically significant indirect effects on educational attainment via GPA and teacher ratings of academic motivation in adolescence, and the smoking PGS also had a small indirect effect via disciplinary problems.

Cumulatively, the random intercept scores for the four academic variables and cigarettes per day accounted for about 50% of the adjusted effects of the smoking (B = -.08, 95% CI: -.10, -.05) and educational attainment (B = .12, 95% CI: .10, .14) PGSs on educational attainment in adulthood. Finally, the smoking (B = -.07, 95% CI: -.11, -.04) and educational attainment (B = .12, 95% CI: .10, .14) PGSs continued to have small but significant direct effects on educational attainment in adulthood, even after adjusting for their overlap, the four adolescent academic variables and cigarettes per day.

Given all these results, we think it is best to frame the paper around the conceptualization of the smoking PGS being related to behavioral disinhibition early in the manuscript. One, that was our original conceptualization of the analysis. Two, we think this is most reasonable interpretation of the results. Three, it is much easier for the reader to understand and interpret the analytic strategy and results if the framing is done at the beginning of the paper. 

4) Interpretations of results regarding associations between PGS and educational attainment: 

“Third, the size and meaningfulness of the effect sizes can be emphasized much more, and conclusions should align with expected impact of the findings.”

“The predictive effect sizes of the two PRS are very small – I find these results not very compatible with the strong conclusions in the summary in the discussion ‘The results provided strong evidence that PGSs for smoking—a measure of genetic influences on behavioral disinhibition—and for educational attainment each predicted educational attainment in adulthood. Most importantly, our analyses demonstrated that genetic influences on behavioral disinhibition provide incremental prediction of educational attainment, even after accounting for a PGS specifically designed to predict educational attainment.’”

We have reviewed the manuscript and find that all our statements are consistent with the data analysis presented. Note that when we state, “The results provide strong evidence that PGSs for smoking and educational attainment each predicted educational attainment in adulthood.”, we do not state that the PGSs exhibit large effect sizes in their association with educational attainment in adulthood. We think the phrase “the evidence is strong” is appropriate to describe the situation wherein a biological measure that can be assessed in childhood predicts a complex adult outcome, even after accounting for several much more proximal covariates (both in terms of age and phenotypic overlap with the outcome). This is especially true for the smoking PGS, because, in theory, the educational attainment PGS should account for all the genetic influences on the educational attainment phenotype that can be measured using a common array of genetic markers (our results indicate this is not the case). Including cigarettes per day in the analysis, a putatively expressed phenotype of the smoking PGS that has a robust association with educational attainment, further increases the rigor and riskiness of the test of the ability of the smoking and education PGS to predict educational attainment. We also devote substantial text in the Discussion to various limitations to the study including the size of the effects, generalizability, and other potentially relevant variables that influence the strength of inferences that can be drawn given the data and analysis. See text below (pp. 23-24): 

Notably, we were only able to account for about one third of the variability in educational attainment. This was in spite of several design strengths including inclusion of relevant academic variables assessed on multiple occasions using multiple informants and several facets of academic adjustment in addition to the two PGSs. Also, the sample was not racially or ethnically diverse, which reduces variability in the United States. Unassessed variables may account for substantial portions of additional variance in educational attainment, such as family attitudes about education and the availability of resources to contribute to obtaining higher levels of education (47). Whether a person pursues advanced education, however, depends on both idiosyncratic and social-structure factors such as availability of job opportunities not requiring additional education, family and partner relationships, specific academic experiences (e.g., satisfying versus dissatisfying), financial constraints, stereotypes about pursuing certain fields of interest, and incentives to return to school after an extended hiatus. Such factors were not well captured in our models. 

The study had other limitations. The PGSs did not identify specific genetic variants that point to biological processes that might account for their associations with educational attainment. Functional genomic information is needed to understand the biological processes accounting for these associations (48, 49). Also, while the hope is that PGSs will eventually have practical value in predicting individual outcomes and informing intervention efforts, this is not yet viable given the small effect sizes. Further, the sample was restricted to people of European ancestry and persons growing up in Minnesota so it is unclear whether the results generalize to other ancestral groups with different allelic frequencies, or societies with different educational systems (e.g., societies with weaker educational infrastructure and fewer opportunities or those with universal access to higher education). Additionally, societal influences related to racial, ethnic, and gender inequities and discrimination in education and cultural values and resources committed to education might moderate genetic influences measured by the PGSs (50). Given substantial overlap between ancestry status and socially defined racial/ethnic status, efforts to improve educational outcomes using PGS approaches have the potential to increase existing disparities if these findings are only applicable to people of European ancestry or culturally defined White people, further prioritizing extending these kinds of studies to diverse ancestry and racial/ethnic groups (51). 

5) Presentation of PGS regression results. 

“Table 2 was a tad confusing, especially the columns relating to the 2 PRS analysis. If there are two PRSs in that analysis, why is there only a single value for each criterion? I might be missing something obvious here, perhaps.”

Coefficients for the 2 PGS models are presented next to the coefficients for the 1 PGS model, to facilitate examining the change in coefficients after adjusting for the overlap in the other PGS. The two PGS results are only provided for the random intercept criterion variables. 

6) Gene-environment correlation. 

“I was not entirely clear how the authors are testing rGE from their into setup or their methods. Academic process and outcome variables are going to be influenced by both genes and environment, most certainly. How is a correlation between e.g., nicotine PRS with academic process variables a test of rGE? Either be clear about the logic or tone it down.”

The text being referred to is merely to acknowledge that the intermediate phenotypes or mediating variables have both genetic and environmental contributions. To the extent we conduct a test of gene-environment correlation, it is to regress the intermediate phenotypes (genetic and environmental variable) on the PGS’s (genetic only variable), so that a significant association would be evidence of a gene-environment correlation. 

7) Possible rater and sex effects.

“Possible rater and sex effects. The authors are using sex as a covariate, and they are also collapsing mom and child reports of educational process. This is reasonable, but the readers might also want to know if there are rater and sex effects. It might be good to redo the models by rater and separately by sex and supplement the information (even though the sex effects models might have power issues, they might also highlight if the effects are particularly strong in one sex).”

These analyses provide a level of detail that are beyond the scope of our aims and so we do not include them in the manuscript. However, we have posted the data to a public server so that those interested can complete these analyses if they choose. 

I confirm that this work is original and has not been published elsewhere, nor is it currently under consideration for publication elsewhere. A pre-print version of this manuscript was posted on 2/3/21 (https://psyarxiv.com/mueqg/) and pre-print version of the revision will be posted following submission of the revision. All authors have no conflicts of interest to disclose. 

Thank you for considering of this manuscript. 

Sincerely,

Brian M. Hicks, Ph.D.

Associate Professor 

Department of Psychiatry

University of Michigan Medical School

---

## [Decision Letter · Decision Letter 1]

15 Jul 2021

Polygenic scores for smoking and educational attainment have Independent Influences on academic success and adjustment in adolescence and educational attainment in adulthood

PONE-D-21-06130R1

Dear Dr. Hicks,

We’re pleased to inform you that your manuscript has been judged scientifically suitable for publication and will be formally accepted for publication once it meets all outstanding technical requirements.

Kind regards,

Edelyn Verona

Academic Editor

PLOS ONE

Additional Editor Comments (optional):

Reviewers' comments:

Reviewer's Responses to Questions

**Comments to the Author**

1. If the authors have adequately addressed your comments raised in a previous round of review and you feel that this manuscript is now acceptable for publication, you may indicate that here to bypass the “Comments to the Author” section, enter your conflict of interest statement in the “Confidential to Editor” section, and submit your "Accept" recommendation.

Reviewer #1: All comments have been addressed

Reviewer #2: All comments have been addressed

2. Is the manuscript technically sound, and do the data support the conclusions?

Reviewer #1: Yes

Reviewer #2: Yes

3. Has the statistical analysis been performed appropriately and rigorously? 

Reviewer #1: Yes

Reviewer #2: Yes

4. Have the authors made all data underlying the findings in their manuscript fully available?

Reviewer #1: Yes

Reviewer #2: Yes

5. Is the manuscript presented in an intelligible fashion and written in standard English?

Reviewer #1: Yes

Reviewer #2: Yes

6. Review Comments to the Author

Reviewer #1: (No Response)

Reviewer #2: I have offered various suggestions to improve the paper. It seems that the authors are determined to stand by their choices and did provide quite a bit of text to explain why they feel no improvements were necessary.

Whereas I don't agree (particularly with regard to the 'strong evidence' statements and really also still about the use of a PRS for smoking as a proxy for dis-inhibition in general) - none of this is sufficiently grave to change my overall opinion that this should be published.

7. PLOS authors have the option to publish the peer review history of their article (what does this mean?). If published, this will include your full peer review and any attached files.

Reviewer #1: No

Reviewer #2: No

---

## [Editor Report · Acceptance letter]

4 Aug 2021

PONE-D-21-06130R1 

Polygenic Scores for Smoking and Educational Attainment have Independent Influences on Academic Success and Adjustment in Adolescence and Educational Attainment in Adulthood 

Dear Dr. Hicks:

I'm pleased to inform you that your manuscript has been deemed suitable for publication in PLOS ONE. Congratulations! Your manuscript is now with our production department. 

Kind regards, 

on behalf of

Dr. Edelyn Verona 

Academic Editor

PLOS ONE